# Release of Melamine and Formaldehyde from Melamine-Formaldehyde Plastic Kitchenware

**DOI:** 10.3390/molecules25163629

**Published:** 2020-08-10

**Authors:** Ingo Ebner, Steffi Haberer, Stefan Sander, Oliver Kappenstein, Andreas Luch, Torsten Bruhn

**Affiliations:** Department of Chemical and Product Safety, German Federal Institute for Risk Assessment, Max-Dohrn-Str. 8-10, 10589 Berlin, Germany; shaberer@gmx.net (S.H.); Stefan.Sander@bfr.bund.de (S.S.); Oliver.Kappenstein@bfr.bund.de (O.K.); Andreas.Luch@bfr.bund.de (A.L.); Torsten.Bruhn@bfr.bund.de (T.B.)

**Keywords:** melamine, formaldehyde, migration, release, food contact material, activation energy, CP/MAS ^13^C-NMR

## Abstract

The release of melamine and formaldehyde from kitchenware made of melamine resins is still a matter of great concern. To investigate the migration and release behavior of the monomers from melamine-based food contact materials into food simulants and food stuffs, cooking spoons were tested under so-called hot plate conditions at 100 °C. Release conditions using the real hot plate conditions with 3% acetic acid were compared with conditions in a conventional migration oven and with a release to deionized water. Furthermore, the kinetics of the release were studied using Arrhenius plots giving an activation energy for the release of melamine of 120 kJ/mol. Finally, a correlation between quality of the resins, specifically the kind of bridges between the monomers, and the release of melamine, was confirmed by CP/MAS ^13^C-NMR measurements of the melamine kitchenware. Obviously, the ratio of methylene bridges and dimethylene ether bridges connecting the melamine monomers during the curing process can be directly correlated with the amount of the monomers released into food.

## 1. Introduction

Polymers produced from the monomers melamine and formaldehyde are used in a variety of food contact materials (FCM). These melamine resins are hard, unbreakable and have a certain degree of thermal resistance, and are thus used for the production of dishes and kitchen utensils. The polymerization is initiated by methylolation of the melamine molecule; further reactions include methylene bridge and methylene–ether bridge formation, leading to a three-dimensional polymeric network. The curing process is well investigated [1] and there are several analytical methods that help to understand the build-up of the 3D-structure during the process. ^13^C and solid-state cross polarization/magic angel spinning (CP/MAS) ^13^C-NMR [2,3,4,5,6] have especially been intensively used to investigate the curing of melamine-formaldehyde (MF) resins. In addition, FTIR studies were performed on MF resins and gave more insight to the crosslinking in the polymer [7,8,9]. However, these methods were never used to find a correlation between release of melamine or formaldehyde into food simulants or foodstuffs and the crosslinking of the resin. Nonetheless, an investigation [10] already stated a correlation between this release and price of the investigated samples; however, this was not confirmed by later studies [11].

Melamine kitchen utensils and tableware were investigated for the release of melamine and formaldehyde to food or food simulants starting in the mid-eighties [12]. In 1992 Martin et al. published a study [13] about melamine tableware on the Philippian market showing the release of melamine and formaldehyde under conditions of 30 min at 95 °C in 4% acetic acid. Lund and Petersen demonstrated in 2006 that significant amounts of melamine and formaldehyde are released from melaware bought on the Danish market to food simulants at higher temperatures (70 °C and 95 °C). Already in 2008 Ingelfinger [14] warned of the impact of melamine food contamination, especially in relation to infant formula that has been contaminated in China, which led to several severely ill or dead children. In the following years, melamine got more into the scientific focus [15,16] and Bradley et al. confirmed that under acidic conditions at temperatures of 70 °C and higher, high amounts of melamine and formaldehyde might migrate or be released into food simulants [17,18]. In the following years, several studies more or less confirmed the earlier results [10,11,19,20].

Usually, melamine is detected by LC-MS/MS or LC-DAD, whereas formaldehyde is determined by LC-DAD or photometric methods, but other analytical techniques are also used [21,22,23].

The release of formaldehyde and melamine from plastics to food or food simulants is still of high concern. As a result of its reassessment of melamine in 2010, the European Food Safety Authority (EFSA) lowered the tolerable daily intake (TDI) from 0.5 to 0.2 mg/kg body weight and day [24]. As a consequence, the specific migration limit (SML) for melamine was lowered in 2011 from 30 mg/kg food or food simulant to 2.5 mg/kg (equivalent to 0.42 mg/dm^2^) by the European Commission [25]. The SML for formaldehyde is set to 15 mg/kg [26] but in a recent opinion of the German Federal Institute for Risk Assessment (BfR) [27] a lowering of the SML to 6 mg/kg was proposed. Melamine products from China are under specific control according to Commission Regulation 284/2011 [28].

For cooking utensils such as spoons, Regulation (EU) number 10/2011 [26] foresees test conditions with temperatures of 100 °C or reflux temperature [29]. Repeated use articles have to be tested three times consecutively, using the third release test result for compliance checking. Foodstuffs with a pH below 4.5 shall be tested with simulant B, e.g., 3% (*w/v*) acetic acid.

In contrast to migration processes controlled by diffusion and/or partitioning alone, release of melamine and formaldehyde from melamine kitchenware increases in several cases with consecutive migration test cycles due to advanced hydrolysis of the polymer. This has been shown in different studies, e.g., by Mannoni et al. [30] and the references therein. Historically, the term “migration” from plastic food contact material was introduced to describe the results of diffusion processes which might be kinetically and/or thermodynamically controlled. Since plastic can also be subject to degradation, all these different processes of release were indiscriminately summarized under the term “migration” [31]. In the context of legislation, the term “migration” is also used in this generalized form [26]. Even if “migration” and “release” are often used interchangeable, for clarity reasons we differentiate in this work between “migration” in its original context of diffusion processes. The term “release” is applied to designate any mechanism of substance transfer from a food contact material and article to food or food simulants.

As binding conditions have an important role for the understanding of the results presented in this work, a brief discussion of those conditions follows here. A melamine resin consists of the monomers melamine and formaldehyde. The chemical structure and number of crosslinks in the polymer are largely defined by the initial molar ratio of these monomers and by pH, temperature and duration of the curing process (Figure 1) [5,32,33].

Building of the crosslinks in the polymer is influenced by the ratio of the monomers and the reaction conditions during the curing process. Melamine **1** and formaldehyde react to methylol melamine **2** (Figure 1). Then **2** can react with another methylol melamine, building dimethylene–ether bridges (in the following only stated as ether bridges) or with a melamine to build a methylene bridge. There are a few possible side reactions at the nitrogen [32], but these occur only in a negligible amount and will thus not be discussed here. In this way a melamine resin with different crosslinks will occur as shown in Figure 2. The ether bridge is the kinetically favored product while the methylene bridge is thermodynamically favored [4,7,8,33].

## 2. Materials and Methods

### 2.1. Samples

Eight different spoons made of melamine resin were purchased in 2010 and 2011 from retail outlets in Germany (see Appendix A). Initially, only spoons 1–3 were purchased and the first experiments were solely done with this set. To have a more general sample pool for later release experiments, a further five spoons were acquired to allow experiments with either all eight spoons or the second set of five spoons. All experiments were done shortly after purchase of the spoons. Before testing, the samples were cleaned by hand with warm soapy water and dried as in domestic use. For repeated testing in contact with foods or simulants, samples were rinsed with water and wiped clean using laboratory tissue paper between exposures.

Food samples were purchased shortly before the experiments in a Berlin supermarket. Apple and sauerkraut juice were used as they were, and plum puree and strained tomatoes were diluted with water (500 g food with 300 mL water).

### 2.2. Chemicals

Melamine (1,3,5-triazine-2,4,6-triamine, C_3_H_6_N_6_, CAS number 108-78-1) with a purity of 99% and the respective isotopically labelled compound, melamine-^13^C_3_ (99,8%, 99% ^13^C_3_, CAS number 1173022-88-2), as internal standards, were purchased from Fluka (Sigma-Aldrich, Munich, Germany).

Acetic acid, HPLC-grade acetonitrile, HPLC-grade methanol and sodium nitrite were purchased from Merck (Darmstadt, Germany). A 3% (*w*/*v*) acetic acid solution was made by filling up 3 g acetic acid to 100 mL with deionized water. Acetylacetone (pentane-2,4-dione, >99%), 98% ammonium acetate and formaldehyde (37.3% *w*/*v*) were obtained from Sigma-Aldrich (Munich, Germany). Deionized water was produced with a Milli-Q system (Merck Millipore, Darmstadt, Germany).

### 2.3. Release Tests

For stove top (hot plate) experiments magnetic stirrers with heating capabilities were used. The release test solution was filled into a 1000 mL beaker, covered with aluminum foil and allowed to reach the designated temperature (either boiling point or at lower temperatures, controlled by external microprocessor controlled contact thermometers) under stirring with PTFE stir bars. After reaching the set point the aluminum foil was cut in such a manner that the spoons could freely rotate. The spoons were put into the release test solution and were set into rotation (100 min^−1^) by means of an electronic stirrer. To maintain comparable conditions throughout all release experiments and a constant volume, any evaporated solution was regularly refilled with water of the same temperature. After the designated time (60 or 120 min) the rotation was stopped, the spoons were taken out of the release test solution, the beaker was again completely covered and the solution was allowed to cool down. The experimental setup is shown in Appendix A and Video S1 of the Appendix A.

Static high temperature tests (conventional migration testing) in an oven were conducted in an incubator KB-240 (Binder, Tuttlingen, Germany). The spoons were placed into a beaker and brought into contact with 3% acetic acid preheated to 100 °C; the beaker was covered with aluminum foil and then the beaker was placed into an oven set at 100 °C. The temperature of the oven, the temperatures of the food simulant and the humidity in the oven were recorded.

The design of the release experiments was not to prove compliance of individual articles, but to recognize general trends for a variety of foods, food simulants and testing conditions. If not stated otherwise, measurements were therefore not replicated to reduce the number of analyses.

As internal standard, ^13^C_3_-melamine was added to a portion of the acidic or aqueous simulant samples, and afterwards they were diluted with acetonitrile 1:19 prior to direct melamine analysis. The food samples were diluted fivefold with deionized water and homogenized using a food blender. Internal standard ^13^C_3_-melamine was added, the sample was centrifuged and an aliquot of the supernatant was diluted with acetonitrile 1:19 prior to direct analysis by LC-MS/MS. The concentration of the internal standard after the dilution steps was 50 ng/µL.

For formaldehyde analysis, acidic or aqueous simulants were mostly used undiluted. Food samples were fivefold diluted with water and centrifuged. The supernatant was derivatized as described in the section “Analytical methods.”

### 2.4. Analytical Methods

All analytical methods (except the NMR investigations) described in the following were performed in a laboratory with ISO 17025 accreditation (German National Reference Laboratory for food contact materials); quantifications of melamine and formaldehyde were performed according to validated and accredited methods.

#### 2.4.1. Melamine

The melamine levels were measured using an HPLC device (LC-20A, Shimadzu, Kyoto, Japan) coupled with an API 4000Q triple-quadrupole mass spectrometer (API 4000QTrap, Applied Biosystems/MDS SCIEX, Concord, ON, Canada). The HPLC column used was a Luna HILIC (3 µm, 200 Å, 150 × 3.0 mm Phenomenex, Aschaffenburg, Germany) eluted with 0.3 mL/min 5 mM ammonium acetate in water (A) and 5 mM ammonium acetate in methanol (B) with a gradient A:B of 5:95 (0–2 min), to 40:60 at 4.9 min (held to 6.0 min), to 5:95 at 7.0 min, equilibration for 3 min and then re-injection. The injection volume was 10 µL and the column was maintained at 40 °C. Electrospray ionization was used in the positive mode with MRM transitions of *m*/*z* 127→85 for quantification and 127→68 for confirmation of melamine. The corresponding MRM transitions of 130→87 and 130→70 were monitored for the labelled internal standard. The turbo ion-spray source was run in the positive mode at a temperature of 550 °C with the following settings: curtain gas, 20; source gas 1, 45; source gas 2, 45; collision activated dissociation (CAD) gas pressure, high; ion spray voltage, 5500. Product ions of m/z 85 and 68 for melamine were obtained using collision energies (CEs) = 27 and 43, respectively, declustering potential (DP) = 46, collision exit potentials (CXPs) = 6 and 12, respectively and entrance potential (EP) = 10. Product ions of *m*/*z* 87 and 70 for internal standard were obtained using CE = 27 and 41, DP = 71, CXP = 6 and 12 and EP = 10. Spiking levels for recovery checks were at 10 and 50 mg/kg. The limit of quantification (LOQ) was dependent on the matrix and the dilution regime used. For undiluted release test solutions it was 20 µg/L.

#### 2.4.2. Formaldehyde

Food simulants were analyzed with a Shimadzu UV-1700 double-beam UV-VIS spectrophotometer, cuvette path length 1 cm, no blank subtraction. All samples of exposed simulant, blank simulant and standards were subjected to a derivatization procedure with acetylacetone and ammonium acetate. The absorption of the resulting complex was measured at 412 nm. Any samples that gave a response at 412 nm outside the calibration range were diluted with 3% (*w*/*v*) acetic acid and re-analyzed. Quantification was achieved by means of external standard calibration using 3% (*w*/*v*) acetic acid fortified with known amounts of formaldehyde. A stock solution of formaldehyde with a concentration of 1 mg/mL was prepared and verified according to the procedure described in the CEN Technical Specification TS 13130 Part 23 (CEN 2004) [35]. This solution was further diluted to give calibration standards of 0.05, 0.1, 0.2, 0.3, 0.5, 1, 2.5, 5, 7.5, 10, 15, 20, 25, 30 mg/L in 3% (*w/v*) acetic acid. The method used was that described in CEN 2004 [35] with slight modifications [36]. The verification of the method was achieved by a successful participation on an interlaboratory comparison organized by the European Reference Laboratory for Food Contact Material in 2011 [37].

Due to the high complexity of foods and beverages, application of HPLC-DAD is better suited than the photometric approach. Therefore, a HPLC-DAD approach after derivatization with acetylacetone and ammonium acetate was used for foods and beverages. Due to the instability of the derivatization product 3,5-diacetyl-1,4-dihydrolutidine, a further oxidation to 3,5-diacetyllutidine was carried out. For the derivatization 1 mL sample was mixed with 0.2 mL derivatization reagent (same as in UV method) in an autosampler vial. The capped vial was mixed and left in a water bath maintained at 60 °C for 15 min. After cooling down on ice the vial was opened, 10 µL of a 10 mg/L sodium nitrite solution and 300 µL acetic acid were added, the vial was re-capped and shaken.

The HPLC system was an Ultimate 3000 (Dionex, Idstein, Germany) equipped with a DAD. The LC column was a Synergy 4µ Polar-RP 80 Å (150 × 2.0 mm Phenomenex, Aschaffenburg, DE) eluted with 0.5 mL/min water (A) and acetonitrile (B) with a gradient A:B of 90:10 (0–1 min), to 55:45 at 8 min (held to 9 min), to 90:10 at 9.3 min and then equilibration for 3 min. The injection volume was 10 µL and the column was maintained at 40 °C. UV detection at 220 nm was used for quantification. The results were corrected by the recovery (95%). Spiking levels for recovery checks were at 0.15, 0.75 and 3.0 mg/kg. The limit of quantification (LOQ) was dependent on the matrix and the dilution regime used. For undiluted release test solutions it was 100 µg/L.

### 2.5. CP/MAS ^13^C-NMR

The solid-state CP/MAS ^13^C-NMR investigations were done with a BRUKER Avance 400 with cross polarizations of 1, 2 and 4 ms. A rotatory frequency of 6.5 kHz with 7 mm rotors was applied with a repetition every 3 s and overall 2048 scans. Decoupling was done using ^1^H-TPPM. To ensure comparability between the measurements, spectra with cross polarizations of 1, 2 and 4 ms were summed; however, an uncertainty of about 15% remained. Nonetheless, relative intensities of the signals are comparable to a very good degree (see Appendix A).

## 3. Results and Discussion

### 3.1. Stirring of Spoons During the Release Test

To examine within-batch variability of the samples, two sets of spoons were tested in triplicate each with a first, second and third exposure to food simulant (3% acetic acid on hot plates). Every experiment was conducted with and without stirring of the spoons. The standard deviation of the release of melamine and formaldehyde from the spoons was between 10% and 20% in the third release step. Without stirring, the deviation was slightly higher and there was a tendency toward higher values compared to experiments with stirring of the spoons. In the first release step a much larger standard deviation was observed with values between 10% and 50% and especially the results from experiments without stirring show the highest deviations (Appendix A). In all later experiments on hot plates, spoons were stirred.

### 3.2. Release into Food Simulants and Food

In accordance with Regulation (EU) No 10/2011, 3% acetic acid was chosen as a commonly used worst case food simulant (food simulant B). To compare the influences of different matrices on the release of melamine and formaldehyde, three melamine spoons were further tested in the four benchmark foodstuffs apple juice, plum puree, sauerkraut juice and strained tomatoes. The test temperature was determined by the boiling point of the samples (~100 °C). A test time of two hours was chosen in accordance with Regulation (EU) No 10/2011 to account for cooking situations lasting longer than one hour. For a better comparison the ratio of the release into foods and the release into the food simulant was calculated and given in percent (Table 1 and Table 2). The tables also show the pH values measured in the third release test solutions of the different samples.

For melamine, release results into foods are comparable with those obtained for release into the food simulant. In the third release test the ratios were between 25% (plum puree) and 127% (sauerkraut juice). In comparison to sauerkraut juice, the food simulant slightly underestimates the release of melamine while it overestimates the values for strained tomatoes or plum puree. Although apple and sauerkraut juice do have the same pH, the release of melamine clearly differs, showing that pH cannot be the only factor influencing the release kinetics, although it has a great impact. In general, all matrices show the same increasing trends from release 1 to 3.

In case of the release of formaldehyde to the different matrices, apple juice and sauerkraut juice show more or less comparable values with the 3% acetic acid values. The formaldehyde results in plum puree and strained tomatoes are somewhat lower. This might be explained by a reaction of the released formaldehyde with the food matrix.

However, overall these experiments indicate that acetic acid is an appropriate simulant for acidic foods at boiling temperatures, clearly corroborating the results of Bradley et al. [17] who showed the suitability of 3% acetic acid for hot fill conditions (2 h@70 °C).

In a second set of experiments, the release under hot plate conditions was compared with a release setting at high temperature in an oven. Additionally, results for a release into water under hot plate conditions were obtained. The same three spoons and five additional ones were tested.

Each spoon was exposed three times to the respective conditions. The experiments on the hot plates were conducted as in the first set of experiments. In the oven tests, the spoons were placed into a beaker filled with 3% acetic acid preheated to 100 °C (beaker with the simulant was preheated, the spoon not). The beaker was covered with aluminum foil and then placed into an oven set at 100 °C. The temperature of the oven, the temperature of the food simulant and the humidity in the oven were recorded (Figure 3).

For verification reasons the oven temperature was documented by two independent thermocouples (1, internal oven control; 2, external verification thermocouple). Both were in good agreement. Before and after each migration test the thermocouple used for measuring the migration temperature was situated freely in the oven recording data similar to the above-mentioned thermocouples. At the beginning of the migration that thermocouple was placed into the beaker with the migration sample. In contrast to the oven temperature, which reached 97 °C (mandatory lowest temperature by CEN/TS 13130-1 [38]) again within three min, the temperature in the food simulant dropped down to a value between 91 and 94 °C and at the same time the humidity rose. The oven was not able to heat the simulant and the spoons within a reasonable time to the temperature range needed. These data clearly show that boiling of the simulant is not reached in an oven adjusted to 100 °C, especially when using non-preheated spoons. Anyway, using an oven at 100 °C is not foreseen by the CEN/TS 13130-1. The results for the third exposure of this second set of experiments are given in Table 3.

Migration into 3% acetic acid in the oven experiments tends to underestimate the melamine release compared to hot plate conditions. This is due to the above-mentioned temperature behavior in the oven and the resulting lower test temperature. For formaldehyde the underestimation is not as strong. This can be explained by the volatility of formaldehyde and the more open design of the hot plate experiments, allowing substantial evaporation. An estimation of the extent of that evaporation is given later. In the oven experiments, evaporation was hindered by an almost complete covering of the beakers, leading to a higher formaldehyde to melamine ratio.

The transfer of melamine into water was substantially lower when compared to 3% acetic acid, thereby demonstrating the influence of the pH on the migration level. This was due to the degradation of the polymer under acidic conditions instead of diffusion controlled migration only. Likewise, the formaldehyde transfer into water was much lower than into the acetic acid. Nevertheless, the ratio of the release into water to that into acetic acid was approximately ten times higher than the same ratio for melamine, indicating a contribution of migration of free formaldehyde in addition to the release by degradation. Obviously, free formaldehyde was not completely extracted from the spoons by the previous two release steps in water.

Comparing the results for the food simulant with the SML of melamine [25] an exceedance by at least a factor of 50 can be found. Furthermore, the SML for formaldehyde [26,27] was exceeded in all cases. That indicates the unsuitability of melamine resins for cooking purposes, confirming the 2011 opinion of the BfR [39]. In addition, degradation is visualized by roughening of the once smooth and shiny surface of the spoons after treatment with acetic acid and with different foods.

The third migration of spoons 4 to 8 into 3% acetic acid on a hot plate was additionally investigated time-dependently. For that reason an aliquot was taken every 30 min and mass fractions of melamine and formaldehyde were measured. The results of these tests are shown in Figure 4.

For all spoons a linear time dependency of the melamine release can be noted. Release values of the spoons 7 and 8 showed no increase from 60 to 90 min but were nevertheless included in the data evaluation. The formaldehyde release, on the other hand, followed a curved line. This curvature can be explained by evaporation of formaldehyde, as already mentioned above. Taking into account a linear release of formaldehyde too, it is possible to estimate the evaporation and release rate by an iterative calculation of formaldehyde release from the polymer and evaporation from the solution (assuming an immediate equal distribution of formaldehyde in the solution) and fitting of the calculated data to the experimentally derived values. The relative evaporation rate was calculated to be 0.83%/min. The total formaldehyde release was derived from the calculated release rate. By dividing the total release by the formaldehyde content in the solution at the end of the experiment a correction factor for calculating the released formaldehyde could be established. Multiplying the formaldehyde transfer after two hours with this correction factor gave the total formaldehyde release. The average correction factor calculated for the formaldehyde release in this experiment was 1.6.

In a further experiment the evaporation of pure formaldehyde solutions was examined and a factor of 1.2 was calculated (data shown in the Appendix A, Appendix A).

### 3.3. Temperature Dependence

Another set of experiments was conducted to establish a correlation between temperature of the experiments and the extent of the monomer release by hydrolysis. To that purpose, already used spoons from the second set of experiments (3% acetic acid, hot plate) were further exposed to boiling 3% acetic acid on hot plates for two hours on the day before the experiments in order to equalize the starting conditions of the polymer surface and to minimize the contribution of the migration of free monomers. The experiments were carried out in duplicate, first in descending order from the highest temperature (100 °C) to the lowest one (40 °C), and second in ascending order. The contact time was one hour. For temperature stability the hot plates were equipped with external microprocessor-controlled contact thermometers. The melamine release was assumed to be linear with time, as shown in the previous experiments (Figure 4A), and the rate constants were calculated according to a kinetic of pseudo-zero-order. An Arrhenius plot was constructed from the release results (mean value, rate constant k in logarithmic values [ln(k)] and the release temperature [1/(RT)]). The Arrhenius plot (inset) and the temperature dependence of the relative release with respect to the highest release values at 100 °C can be found in Figure 5.

Temperature dependence of the release of melamine strictly obeys the Arrhenius equation. The calculated activation energies range between 120 and 130 kJ/mol. For comparison, a curve representing an activation energy of 80 kJ/mol was drawn. This energy was derived from the original version of regulation (EU) number 10/2011 where it represented the worst case activation energy for migration processes [26].

In addition, the obtained melamine releases resemble an average of 96% (81% to 133%) of those measured in the third release test shown in Table 3 (3% acetic acid, hot plate conditions), clearly confirming the constancy of the release after the third release test.

In a study performed by Chien et al. [10] different melamine cups were investigated at temperatures between room temperature and 90 °C. In Figure 6 the relative release results are compared with the curve representing an activation energy of 120 kJ/mol obtained from our investigations. Despite the very different experimental setup, a good correlation was obtained at temperatures above 60 °C (see dashed reference lines in the inset of Figure 6). Below 60 °C the results from the literature work show higher values. This might be explained by a substantial contribution of migration in addition to the release by degradation in the investigations of Chien et al. It can be stated that the average temperature dependence of Chien et al. consists of two slopes (inset in Figure 6), one from room temperature to approximately 50 °C, most probably with a significant contribution from migration of free monomers, and the other above 50 °C, mainly dominated by degradation.

### 3.4. Ratio of Formaldehyde/Melamine

As mentioned above, release of melamine and formaldehyde into 3% acetic acid at boiling conditions is mostly influenced by a degradation of the polymer (after the free monomers from synthesis and curing processes have been depleted by migration). Therefore, release values of these should reflect the monomer contribution to the polymer. In Table 4 the test results from the hot plate experiments in 3% acetic acid are presented as molar ratios of formaldehyde to melamine (F:M ratio). Formaldehyde values were corrected by a factor of 1.4, the average of the before-established two factors (see Section 3.2).

In all cases, the F:M ratios decreased in the consecutive release tests. For the 3% acetic acid and hot plate conditions, the F:M ratios determined by the second and third release tests (Table 4) differed only slightly, thereby indicating that a near constant degradation rate had been reached and migration of free formaldehyde was neglectable. The data on the other experimental conditions show a more substantial contribution of free formaldehyde migration expressed by much higher F:M ratios. However, for the release tests conducted in an oven the F:M ratios converged to the ones on a hot plate in the third test solution.

Data of the third release tests into 3% acetic acid on a hot plate were used to further study the ratio of the monomers. Figure 7 shows the dependence of the melamine release on the molar ratio of melamine and formaldehyde.

The F:M ratio ranged between 1.7 and 3.0. This resembles molar ratios typically used for melamine resin production [4,6] and is in the same range as in the so-called import group described in the work of Mannoni et al. [30]. However, in the latter study kitchen also utensils with much higher F:M ratios were found (up to 5.2). The area-related melamine release is used in our work as an expression of the decomposition of the polymer. A correlation of the melamine release with the F:M ratio was observed; however, values for spoon 7 seems to be an outlier for an unknown reason, maybe because of different fillers or an unusually high amount of free formaldehyde in the polymer.

The release of melamine depends on the degradation of the ether or the methylene bridges. The number of these bridges is determined by the production conditions (see Introduction). A higher F:M ratio is associated with a higher number of ether bridges, either because they were initially kinetically favorably formed or because they did not further react to methylene bridges due to lower curing temperatures [1,9]. Under acidic conditions the back reaction of the ether bridges is also kinetically favored [5,9,40]; a higher number of ether bridges leads thus to a higher hydrolysis rate. Moreover, cleavage of the ether bridge yields two formaldehyde monomers in contrast to one monomer for that of a methylene bridge. The more ether bridges occurring in the melamine resins, the more release of formaldehyde monomer expected to be determined by a higher F:M ratio. Additionally, this led to a higher melamine release, as depicted in Figure 7.

Here, a clear influence of the monomer composition on the release of monomers from the cured product is indicated, which is in contrast to the results of Mannoni et al. The ratio of ether and methylene bridges is a quality attribute of the resin and can be monitored by NMR or FTIR investigations [9]. Therefore, we performed NMR investigations to further confirm the F:M ratio results.

### 3.5. NMR Investigations

We performed solid-state CP/MAS ^13^C-NMR measurements of the spoons to investigate if there was a correlation between the release of melamine and formaldehyde and the ratio of methylene or ether bridges in the samples. One has to keep in mind that not only was the melamine resin detected in the NMR, but the filler used too. A commonly used filler for melamine resins is cellulose. The NMR measurements (Figure 8) showed that in all spoons cellulose had been used, and the according signals were attributed by comparison with values from the literature [3].

^13^C-NMR signals of the filler cellulose are mainly located in the range from 60 to 110 ppm, with the best resolved signal for cellulose C1 at 105 ppm. As the amount of cellulose used as filler was different in all spoons, the spectra in Figure 8 were scaled to the area of the signal of C1. The most important signals of the melamine resin for this study are located at 168 ppm (carbons of the triazin ring system) and at 49 ppm (methylene bridge) [2,4,5,32]. The methylol signals at ~72 ppm [6,34] and the ether bridge carbons at 66 ppm cannot be resolved from the C2,3,5 and C6 signals of the cellulose.

Due to the scaling to the area of the C1 signal in Figure 8, one can conclude from the intensities of the triazine carbon signals of the spoons that different ratios of melamine resin to filler were used. To confirm that the amount of filler has no influence on the release, we checked whether the cellulose content had an effect on the measured melamine release of the spoons but no correlations were found (data shown in the Appendix A, Appendix A). Therefore, the focus was put on the signals of the methylene bridges in the melamine resin. As mentioned above, the level of the methylene bridges is important and to quantitize it the NMR spectra of the spoons were scaled to the signal intensities of the triazine carbons. The more intense the signals of the methylene bridge are, the higher the number of these more stable crosslinks should be (Figure 9).

Next, a direct/linear correlation between the intensity of the methylene bridge signals and the release of melamine was observed; the higher the intensity the lower the release of melamine. The NMR results and the data from the F:M ratio investigations gave a clear correlation between melamine/formaldehyde release and the quality of the resins (Figure 10). Obviously, higher quality (according to the release of the monomers) can be obtained by choosing conditions during the curing process of the resins that lead to higher amounts of methylene bridges as crosslinks (appropriate pH, low F:M ratio). In contrast to the results presented in Figure 7 wherein the dependence on the crosslinking conditions is only indirectly depicted and might be influenced by an excess of free formaldehyde, here a direct correlation of melamine release to the crosslinking conditions can be drawn. From the combined investigations (F:M ratio and NMR) conclusions on the reasons of a higher melamine release under the tested acidic conditions can be drawn. The more ether bridges and the fewer methylene bridges occurring in the resin, the more migration of monomers, and thus the suitability of the melamine resin as food contact material is lower.

## 4. Conclusions

The comparison of different food matrices with 3% acetic acid clearly shows the suitability of acetic acid as a food simulant for use in release testing of melamine kitchen utensils at boiling temperatures, complementing earlier results of Bradley et al. [17] for hot fill conditions. As keeping to the specified temperature range in the release test is critical to obtaining comparable results, different conditions and experiments were tested. It was demonstrated that stirring and the use of hot plate conditions are essential for precise and comparable results in case of testing of melamine resin spoons or comparable kitchenware. The conventional oven migration conditions without preheating failed to reproduce the values from the hot plate conditions, especially because the selected oven temperature of 100 °C was not sufficient to maintain the correct release conditions.

A deeper investigation on the kinetics of the melamine release into 3% acetic acid showed a clear obeyance of the Arrhenius equation with calculated activation energies of about 120 kJ/mol. More important was the correlation found between the F:M molar ratio and the melamine release. The correlation was corroborated by solid state CP/MAS ^13^C-NMR investigations, which showed that the quality of the resin—defined by the ratio of ether and methylene bridges in the cured material—is responsible for the amount of the release of melamine and formaldehyde. These results should have an important impact on GMP procedures during the production of melamine resins. As it is well known how to influence the curing process of the resin, it should be possible to further minimize the release of melamine and formaldehyde from melaware kitchen utensils.

## Figures and Tables

**Figure 1 molecules-25-03629-f001:**
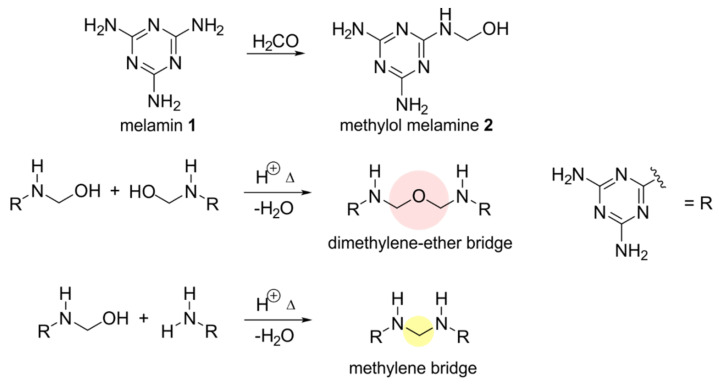
Schematic representation of the initial reactions of the synthesis of melamine resins from melamine and formaldehyde (simplified).

**Figure 2 molecules-25-03629-f002:**
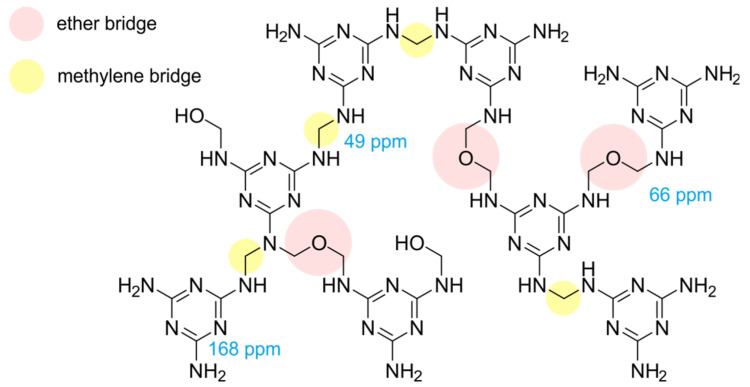
Possible crosslinks in a melamine resin, ether or methylene bridges, with selected ^13^C-NMR chemical shifts taken from the literature [5,6,34].

**Figure 3 molecules-25-03629-f003:**
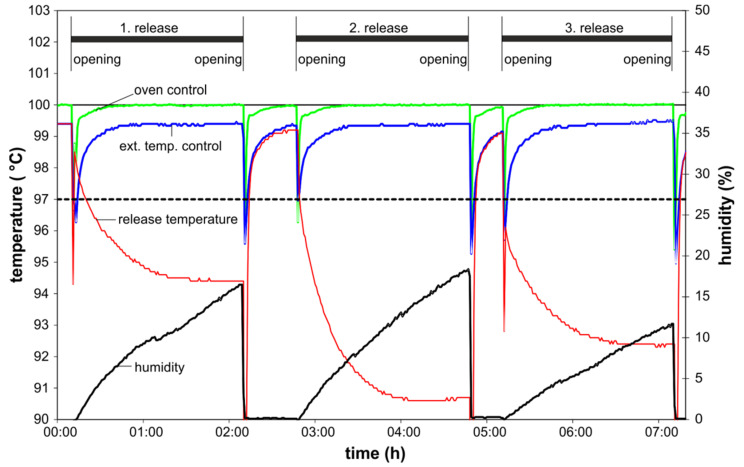
Temperature and humidity in three subsequent release tests in an oven; the black dashed line shows the lower temperature limit for hot plate conditions. Internal oven control: green curve, external verification thermocouple (ext. temp. control); blue curve, humidity: black curve, thermocouple for test solution temperature: red curve.

**Figure 4 molecules-25-03629-f004:**
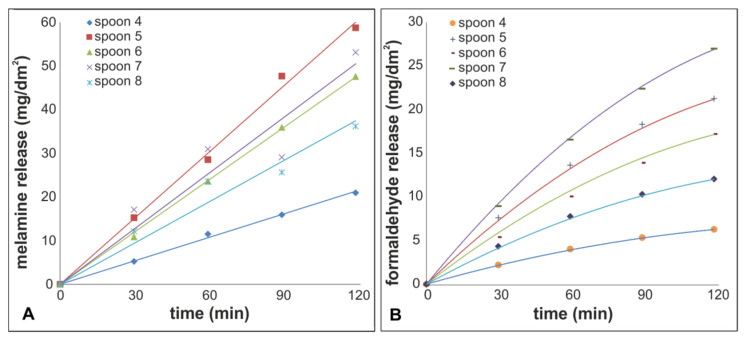
Time dependent measurement of the third release (3% acetic acid, hot plate, stirring), (**A**) melamine, solid lines: linear fit of data; (**B**) formaldehyde, solid curves: formaldehyde in solution, evaporation losses calculated with an evaporation rate of 0.83%/min

**Figure 5 molecules-25-03629-f005:**
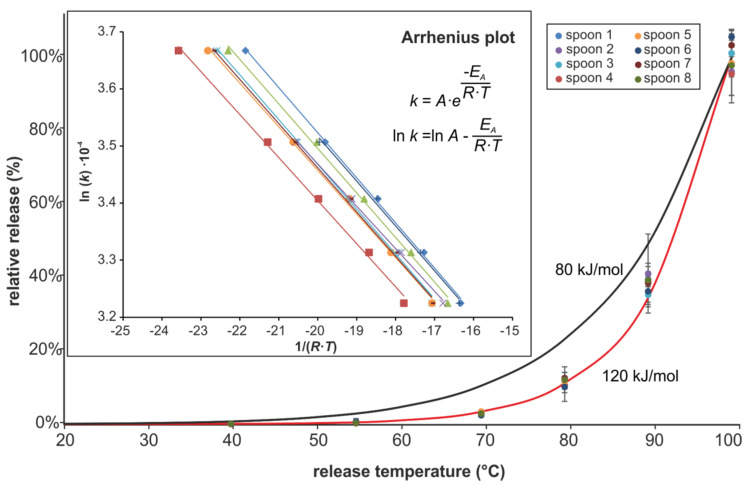
Temperature dependence of melamine release (red line: calculated values for an activation energy of 120 kJ/mol, black line: calculated values for an activation energy of 80 kJ/mol); error bars indicate the range of the averaged duplicates. Inset: Arrhenius plots.

**Figure 6 molecules-25-03629-f006:**
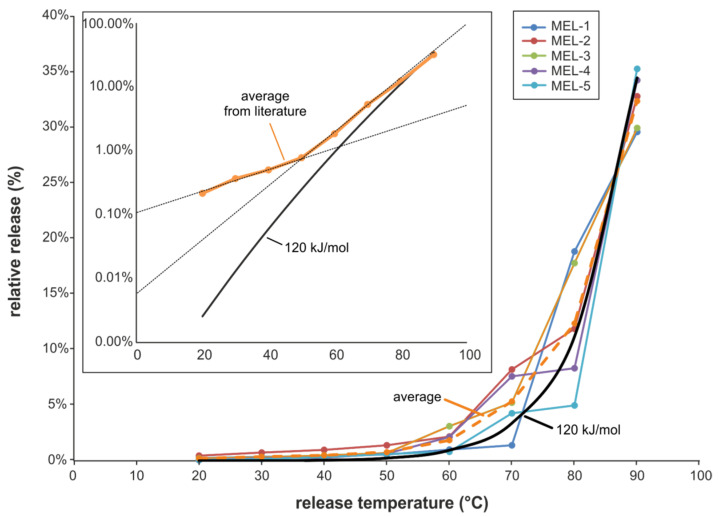
Comparison of literature data [10] on temperature dependence of melamine migration/decomposition (5 different cups, MEL-1 to MEL-5) with the results of this work; dashed broad line (orange): average of the published data, solid broad line (black): this work, calculated values for an activation energy of 120 kJ/mol. Inset: logarithmic scale.

**Figure 7 molecules-25-03629-f007:**
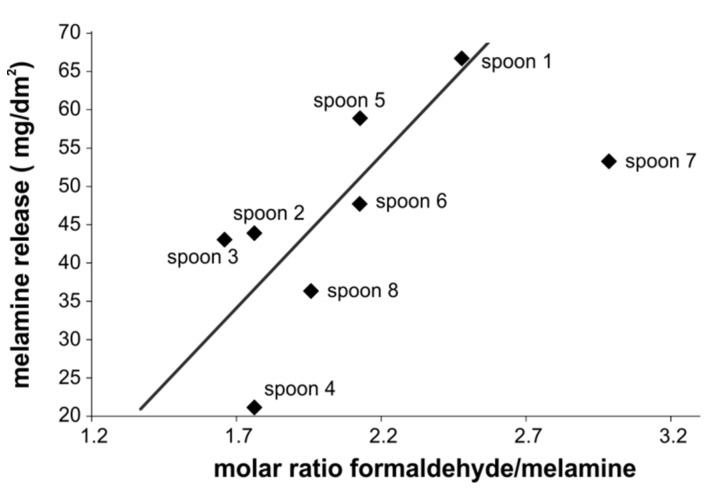
Dependence of melamine release of the molar ratio of formaldehyde to melamine (formaldehyde results corrected by a factor of 1.4, results from the 3rd release test); spoon 7 is treated as an outlier.

**Figure 8 molecules-25-03629-f008:**
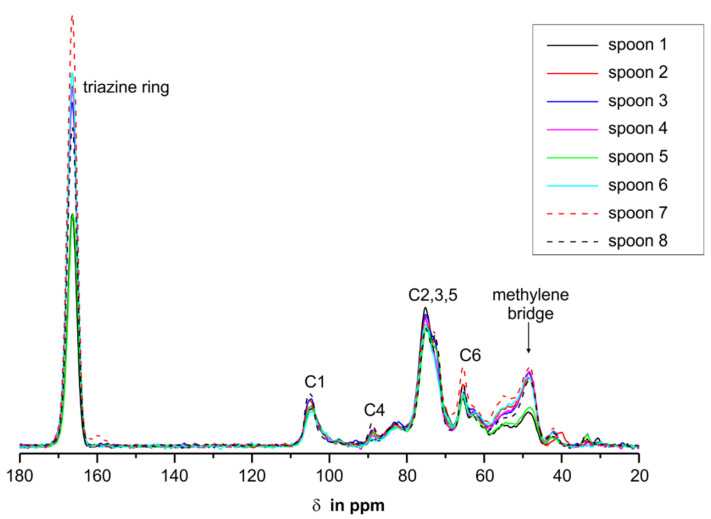
Overlay CP/MAS ^13^C-NMR spectra of the spoons 1–8. For better comparability, the spectra are scaled to the area of the signal of the cellulose C1.

**Figure 9 molecules-25-03629-f009:**
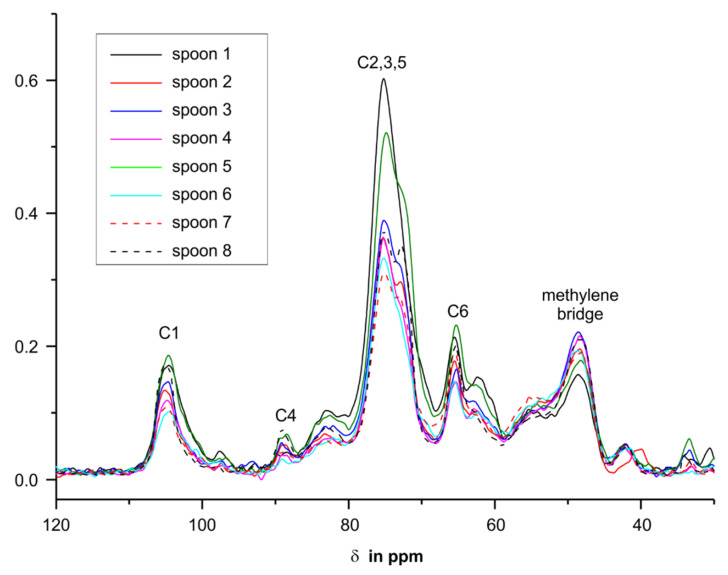
Section of the CP/MAS ^13^C-NMR spectra of the spoons; all spectra are scaled to the intensity of the signal for triazine ring carbons at 168 ppm.

**Figure 10 molecules-25-03629-f010:**
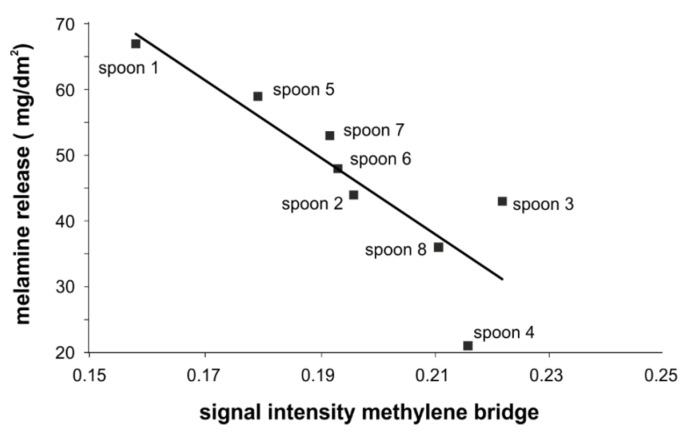
Dependence of the melamine release (3rd release) on the number of methylene bridges in the resin, solid line: fit of data.

**Table 1 molecules-25-03629-t001:** Release of melamine (mg/dm^2^) into acidic foods at hot plate conditions (2h@100 °C) and the ratio (%) of that release compared to 3% acetic acid.

	3% Acetic Acid	Apple Juice		Plum Puree		Sauerkraut Juice		Strained Tomatoes	
pH	2.7	3.5		3.4		3.5		4.2	
spoon 1									
1. release	20	30	147%	24	119%	45	221%	12	60%
2. release	58	40	69%	16	28%	75	130%	33	57%
3. release	67	42	64%	16	25%	72	108%	48	73%
spoon 2									
1. release	12	25	206%	10	79%	36	288%	8	65%
2. release	39	32	82%	16	41%	50	129%	26	68%
3. release	44	23	53%	16	36%	49	112%	32	74%
spoon 3									
1. release	17	17	100%	13	78%	46	275%	13	77%
2. release	46	30	66%	20	44%	65	140%	32	68%
3. release	43	34	79%	19	45%	54	127%	41	97%

**Table 2 molecules-25-03629-t002:** Release of formaldehyde (mg/dm^2^) into acidic foods in hot plate conditions (2 h@100 °C) and the ratio (%) of that release compared to 3% acetic acid.

	3% Acetic Acid	Apple Juice		Plum Puree		Sauerkraut Juice		Strained Tomatoes	
pH	2.7	3.5		3.4		3.5		4.2	
spoon 1									
1. release	24	18	75%	5	20%	26	109%	2	10%
2. release	24	15	62%	3	10%	26	110%	3	14%
3. release	28	16	57%	2	7%	25	88%	7	23%
spoon 2									
1. release	9	9	101%	2	20%	12	141%	2	20%
2. release	12	8	65%	2	14%	12	95%	2	15%
3. release	13	6	49%	2	12%	12	93%	4	27%
spoon 3									
1. release	8	7	79%	2	22%	14	174%	2	27%
2. release	12	8	61%	2	15%	15	118%	3	27%
3. release	12	8	67%	2	14%	13	108%	3	28%

**Table 3 molecules-25-03629-t003:** Results of the third release of melamine and formaldehyde (mg/dm^2^) into food simulants under different migration conditions and the ratio (%) of that migration compared to 3% acetic acid on a hot plate (spoons 1–3 are of the same type as in the first experiments).

	3% Acetic Acid		3% Acetic Acid		Deionized Water	
Test Conditions	Hot Plate		Oven ^a^		Hot Plate	
	Melamine	Formaldehyde	Melamine	Formaldehyde	Melamine	Formaldehyde
spoon 1	67	28	32 (47%)	21 (75%)	1.5 (2%)	6.9 (25%)
spoon 2	44	13	28 (65%)	12 (92%)	0.4 (1%)	2.6 (20%)
spoon 3	43	12	26 (61%)	11 (90%)	0.8 (2%)	1.8 (15%)
spoon 4	21	6.3	16 (76%)	6.8 (108%)	0.3 (2%)	1.2 (20%)
spoon 5	59	21	51 (86%)	26 (121%)	1.3 (2%)	5.4 (26%)
spoon 6	48	17	24 (50%)	11 (64%)	0.5 (1%)	2.0 (12%)
spoon 7	53	27	32 (61%)	18 (68%)	0.4 (1%)	4.3 (16%)
spoon 8	36	12	38 (106%)	15 (124%)	1.4 (4%)	2.6 (21%)

^a^ Release in an oven set to 100 °C, temperature in simulant dropped within 1 h to 91–95 °C.

**Table 4 molecules-25-03629-t004:** Molar ratio of formaldehyde to melamine in release test solutions (formaldehyde results on hot plates corrected by a factor of 1.4).

Test Conditions	3% Acetic Acid			3% Acetic Acid			Deionized Water		
	Hot Plate			Oven ^a^			Hot Plate		
Release Test Step	1	2	3	1	2	3	1	2	3
spoon 1	6.9	2.4	2.5	13	3.8	2.8	42	34	27
spoon 2	4.1	1.9	1.8	6.0	2.2	1.8	90	53	38
spoon 3	2.9	1.6	1.7	9.2	2.4	1.7	60	27	13
spoon 4	7.8	2.3	1.8	6.7	2.2	1.8	52	40	21
spoon 5	6.3	2.6	2.1	8.8	2.7	2.1	86	48	25
spoon 6	6.1	2.5	2.1	7.7	2.4	1.9	83	57	25
spoon 7	4.3	3.4	3.0	10	3.0	2.4	140	102	66
spoon 8	3.5	2.0	2.0	4.0	1.8	1.6	27	19	11

^a^ Release in an oven set at 100 °C, temperature in simulant dropped within 1 h to 91–95 °C.

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
