# Peer review of "Release of Melamine and Formaldehyde from Melamine-Formaldehyde Plastic Kitchenware"

_molecules, 2020, doi:10.3390/molecules25163629_

Round 1

Reviewer 1 Report

The paper “Release of melamine and formaldehyde from melamine-formaldehyde plastic kitchenware” shows a work with a limited degree of novelty and that must be considerably improved before its acceptation for publication in MOLECULES.

A detailed list of items that should be revised are as follows:

  • Spoons were purchased in 2010 and 2011. The study here presented was carried out several years ago and the results are published now, or spoons have been stored for a long time before a recent analysis? In this case, a considerable aging could occur. Please, explain.
  • Aluminium foil is degraded by direct contact or vapour of acidic solutions. Has this been taken into account?
  • The closure of beakers with aluminium foil is not tight, mainly (video in supplementary material) when mechanical agitation is carried out. Therefore, a considerable amount of liquid is lost due to evaporation. There is no reference along the paper about the amount lost (easily assessed by weighting liquid before and after the tests). Before measuring, the lost volume should be considered for correction in concentrations released or readjusted to the original ones to avoid correction factors.
  • Line 113. “If not stated otherwise measurements were not replicated to reduce the amount of analyses”. This limits the validity of most of data. In addition, several times “duplicate” or “triplicate” is mentioned and considered for calculation of deviations. This must be fully revised, because it is difficult to consider results in a reasonable manner if a common criterion is not followed.
  • In general, discussion of results is weak and must be improved. For instance, in line 156, a simple sentence invalidates the UV-VIS procedure.
  • It is clear that for confidentiality purposes, sample coding is different from figure showing the selected spoons. However, what is the reason for selecting three (line 193 and ss, tables 1&2)? The most representative? The most abundant? The most problematic?
  • When calculating area of spoons, and according to data of table S1, the whole spoons have been considered, but only a part of them have been in contact with liquid. In the case of dynamic test, this part can be also be variable. The calculated results, are corrected to consider only the area in contact with foodstuffs/acetic acid or, the complete spoon? Please, revise, because actual results could be considerably higher than those here reported.
  • Comments such as “comparable”, “similar”, “good correlation” are quite arbitrary, since no quantitative data are presented. In addition discussion about results presented in tables are not accurate (e.g. higher concentration at third test is mentioned as a general rule, but numerous exceptions can be seen).
  • Standard deviations are in general too high, thus indicating a low homogeneity of the material used for the same kind of spoon or low robustness of tests.
  • In figure 2, explanation of strange points for melamine is very short. No correlation equations are provided. The calculation of correction factor for formaldehyde is erroneous (0.83% per minute means a full evaporation after 120 min, see previous comment about evaporation of liquid before). In addition, the assumption of an averaged value (1.4), taking into account the additional experiment, for further calculations is also arbitrary in order to get a better fitting.
  • In figure 6, spoon 7 is considered as outlier, whereas in figure 10, spoons 3&4 are not (at least not mentioned). Again, no correlation equation is given, so the accuracy cannot be satisfactorily known.

By all these reasons, I cannot recommend the acceptation for publication of the manuscript in Molecules in its present form.

Reviewer 2 Report

This paper addresses a current problem of the toxic compound release from the plastic kitchenware to food. In the introduction the authors consider recent literature on the melamine and formaldehyde release to food and food simulants, provide the correct statement of the problem and substantiate the analytical methods used. The experimental procedures are described in detail and allow one to reproduce the experiments. The results include a comprehensive comparison of melamine and formaldehyde release from melamine spoons into 3% acetic acid and acidic foods performed at different conditions, including time-dependent and temperature-dependent studies. Moreover, the correlation between the formaldehyde to melamine molar ratio and the melamine release has been established using CP/MAS 13C NMR. The results obtained provide useful recommendations for modification of the procedure of  melamine resin production allowing to minimize the release of the toxic monomers into food. The paper is well written, and the conclusions are fully supported by the data obtained, so it can be published without any serious revision, except the minor corrections: in Fig. 3 and Fig. 4 the axis labels should be written in English (temperature, Arrhenius plot).  

Reviewer 3 Report

This paper presents aims to study the release of melamine and formaldehyde from kitchenware made of melamine resins. The paper is useful to improve the existing test methods, because different conditions and experiments were tested. The effect of several parameters, as temperature range on melamine and formaldehyde release is studied.

In the state of the art, the production of melamine resins, as well as the synthesis reactions and type of links formed during reaction and presented in Figure 5 should be focused. The curing reaction and the crosslinking should also be focused in the introduction section and not in results (Figure 7).

In the materials and methods section, the analysis methods of formaldehyde is not well described. The derivatization of formaldehyde using acetylacetone normally uses ammonium acetate, but there is no mention to ammonium acetate. Regarding the determination of formaldehyde no information is provided how the formaldehyde standard solution was prepared. This solutions is needed for UV/VIS calibration curve. The determination of formaldehyde to melamine molar ratio in the release solutions is not specified.

In the results section, there is a relevant discussions about the effect of test methods parameters, mostly in case of temperature dependence of melamine release using an Arrhenius plot.

The results of the effect of the molar ratio of formaldehyde / melamine on melamine release are not well explained. Why the release of melamine increases with the F/M ratio in test solutions. It could be interesting to compare the F/M ratio of the final resin to produce the spoons with the release of melamine.

In case of NMR investigations there are some doubts about the obtention of quantitative results from CP/MAS 13C NMR spectra regarding the functional groups as methylene or methylene ether bridges. However, the obtained results seems to indicate a good correlation between melamine release from spoons and the intensity of the bands assigned to methylene bridges.

So, I proposed the acceptance of the paper with major revisions.

Round 2

Reviewer 1 Report

Authors have more or less answered the suggestions made in the first revision. Despite well reasoned, I still consider that there are tools (ANOVA, t-test, F-test, p value...) that could be applied for a rigurous comparison of results from a pure statistical point of view. Nevertheless, if the other reviewers consider that the paper can be accepted for publication in Molecules, I have no objection.